# Quantifying the redundancy between prosody and text

**Lukas Wolf**[ʕ]    **Tiago Pimentel**[ð,ʕ]    **Evelina Fedorenko**[ɱ]    **Ryan Cotterell**[ʕ]
**Alex Warstadt**[ʕ]    **Ethan Gotlieb Wilcox**[ʕ]    **Tamar I. Regev**[ɱ]

[ʕ]ETH Zürich    [ɱ]MIT    [ð]University of Cambridge

{wolflu, ryan.cotterell, warstadt, ethan.wilcox}@ethz.ch
tp472@cam.ac.uk    {evelina9, tamarr}@mit.edu

## Abstract

Prosody—the suprasegmental component of speech, including pitch, loudness, and tempo—carries critical aspects of meaning. However, the relationship between the information conveyed by prosody vs. by the words themselves remains poorly understood. We use large language models (LLMs) to estimate how much information is redundant between prosody and the words themselves. Using a large spoken corpus of English audiobooks, we extract prosodic features aligned to individual words and test how well they can be predicted from LLM embeddings, compared to non-contextual word embeddings. We find a high degree of redundancy between the information carried by the words and prosodic information across several prosodic features, including intensity, duration, pauses, and pitch contours. Furthermore, a word's prosodic information is redundant with both the word itself and the context preceding as well as following it. Still, we observe that prosodic features can not be fully predicted from text, suggesting that prosody carries information above and beyond the words. Along with this paper, we release a general-purpose data processing pipeline for quantifying the relationship between linguistic information and extra-linguistic features.

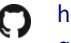 https://github.com/lu-wo/
quantifying-redundancy

## 1 Introduction

In human communication, information flows between interlocutors along multiple channels. Apart from linguistic information conveyed by the text[1] itself, meaningful content is also transmitted through facial, hand, and body gestures as well as through prosody. In spoken language, prosody consists of the acoustic features of the speech signal such as energy, duration, and fundamental frequency ($f_0$).[2] It carries both linguistic information, such as lexical stress, and discourse function, e.g., extra-linguistic information, such as social and emotional cues (Cole, 2015; Ward and Levow, 2021; Hellbernd and Sammler, 2016).

Previous studies have suggested partial redundancy between the acoustic features of prosody and linguistic information. For instance, several prosodic features, e.g., pitch and duration, correlate with the predictability of a word in context (Aylett and Turk, 2006; Seyfarth, 2014; Malisz et al., 2018; Tang and Shaw, 2021; Pimentel et al., 2021) and can mark the location and type of focus in a sentence, emphasizing which information is novel (Breen et al., 2010). These studies suggest that information signaled by prosody is redundant with information from the preceding text, arguably implying that prosody is largely used to direct the interlocutor's attention. However, in some cases, prosody seems essential to transmit unique information, like marking irony, resolving ambiguous syntactic attachments, or transforming a statement into a question. Therefore, characterizing the relationship between the information conveyed by prosody and by words is essential for understanding the role of prosody in human communication.

Our paper quantifies the redundancy between prosody and text using information-theoretic techniques. We operationalize redundancy as *mutual information* (Shannon, 1948, 1951), an information-theoretic measure that quantifies the amount of information (in units such as bits) obtained about one random variable (e.g., prosody) by observing the other random variable (e.g., text). We build upon existing work in the emerging field of information-theoretic linguistics (Brown et al., 1992; Smith and Levy, 2013; Wilcox et al., 2020; Shain et al., 2022). Previous research has applied information theory to analyze various facets of language such as word lengths, ambiguity, and syntax (Piantadosi et al., 2011, 2012; Futrell et al.,

---

[1]We use the word *text* to refer to all the linguistic information that is expressed in words (i.e., segmental information). We are not specifically interested in written texts; for a spoken signal, the text would be the information left behind after the prosodic information has been removed.

[2]The fundamental frequency is defined as the lowest frequency of a periodic waveform, and it gives rise to the perception of pitch.

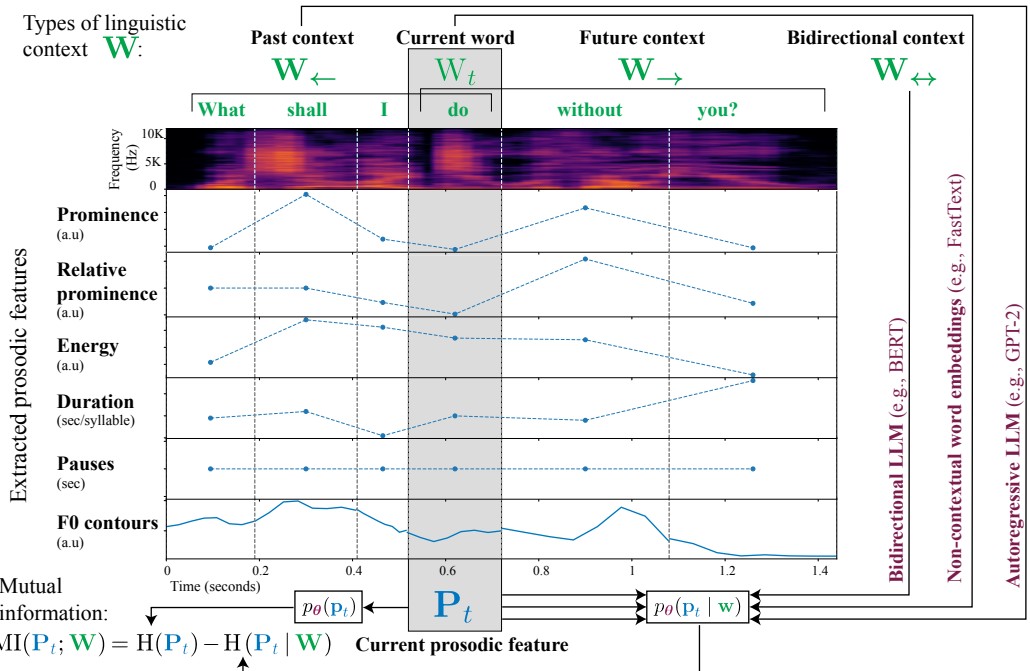

Figure 1: Illustration of our pipeline for estimating the mutual information between prosody and text. The pipeline begins with the extraction of word-level prosodic features. For each type of linguistic context $\mathbf{W}$, we use a corresponding language model to estimate $p(\mathbf{P}_t \mid \mathbf{W})$ by learning a mapping from model embeddings to the prosodic feature. We further estimate the probability distribution of the prosodic feature $p(\mathbf{P}_t)$. We use these probabilities to estimate the entropy and conditional entropy of the prosodic feature. Finally, employing the standard decomposition of mutual information, we arrive at an estimate by utilizing the two computed entropies. See Sections 2-4 for further detail. a.u stands for arbitrary units when prosodic features are $z$-scored.

2019; Meylan and Griffiths, 2021; Levshina, 2022; Pimentel et al., 2023). Yet there is relatively little work specifically characterizing the high-level statistical relationships between prosody and text.

In this study, we employ large language models (LLMs) to estimate the uncertainty about prosody given text, and further compute the mutual information between prosody and text. We work with a corpus of English audiobooks, align the transcript to the speech signal, and extract for each individual word a set of prosody-relevant acoustic features, namely intensity, duration, pitch curves, prominence (a composite measure of the intensity, duration and pitch Suni et al. 2017; Talman et al. 2019) as well as any pausal information that may occur between words. We then finetune a variety of powerful, openly available LLMs to predict the prosodic cues aligned to each word (Figure 8). By relying on different LLMs to make these predictions —either non-contextual,[3] autoregressive, or bidirectional—we measure the redundancy

between prosody, and either the current word, its preceding context, or both its preceding and following context, respectively. As an additional contribution, we publicly release the code we developed to implement our data-processing pipeline. This code and the estimation tools we introduce here are general enough to be applied to other studies on prosody and text, as well as interactions of other communication channels with text.

## 2 Redundancy as mutual information

Our objective is to estimate the redundancy between a text and its accompanying prosodic features. As a first step, we define the notation we will use in this section. Let $\Sigma$ be an alphabet, a finite, non-empty set of symbols. We will consider a **text** to be a string, i.e., an element of $\Sigma^*$, and we will denote it as $\mathbf{w}$. With $\mathbf{W}$, we denote a text-valued random variable. In the experimental section of this work, we will encounter text that is either one word or several sentences totaling no more than 100 words. We further consider **prosody** to be a real-valued vector $\mathbf{p}_t \in \mathbb{R}^d$, and we denote a prosody-valued random variable as $\mathbf{P}_t$.

---

[3]As we explain later, these non-contextual models use multi-layer perceptrons on top of non-contextualized word embeddings.

The subscript $t$ marks the distinguished position of the word in a text whose prosody we wish to investigate. In contrast to $\mathbf{P}_t$, the portion of text that $\mathbf{W}$ corresponds to is currently under-specified. As detailed later in §3, we consider text either before, after, or at position $t$ (see Figure 1).

We now offer a quantitative treatment of the redundancy between prosody and text, operationalizing it as the mutual information:[4]

$$\text{MI}(\mathbf{P}_t; \mathbf{W}) = \qquad\qquad (1)$$
$$\sum_{\mathbf{w} \in \Sigma^*} \int_{\mathbb{R}^d} p(\mathbf{p}_t, \mathbf{w}) \log \frac{p(\mathbf{p}_t, \mathbf{w})}{p(\mathbf{p}_t)\, p(\mathbf{w})} \mathrm{d}\mathbf{p}_t$$

We next discuss a few properties of this mutual information, and how we estimate it.

## 2.1 Estimation challenges

Our mutual information of interest, in Eq. (2), is defined as a function of a discrete ($\Sigma^*$) and a continuous ($\mathbb{R}^d$) random variable. Unlike in the case where the random variables are both discrete or both continuous, the mixed-pair case does not inherently obey many properties one associates with mutual information (Gao et al., 2017), and is therefore challenging to estimate. For instance, it is generally not guaranteed that:

$$\text{MI}(\mathbf{P}_t; \mathbf{W}) = \text{H}(\mathbf{P}_t) - \text{H}(\mathbf{P}_t \mid \mathbf{W}) \qquad (2)$$

which many estimation methods including our own, rely on. However, Beknazaryan et al. (2019) give a simple sufficient condition for the decomposition in Eq. (2) to hold, which they term the **good mixed-pair assumption**.[5] Under this assumption, which we suggest is reasonable to assume in our case where prosodic features are well-behaved and we *can* decompose the mutual information as shown in Eq. (2). For more details on the challenges of mixed-pair mutual information estimation we refer to App. A.1.

Another challenge for estimating mutual information in our case is due to the nature of $\mathbf{W}$. Several mixed-pair estimation methods, including Beknazaryan et al. (2019), may not apply to our case because they require estimating $\text{H}(\mathbf{P}_t \mid \mathbf{W} = \mathbf{w})$ for each given value of $\mathbf{W}$ *separately*; see App. A for more details. However, we use text from books and their spoken counterparts

to estimate the entropy of $\mathbf{P}_t$ given $\mathbf{W}$ (§4.1). Each sentence in these books has a *single* recording, and thus, we only have a single measurement of $\mathbf{p}_t$ for each unique value of $\mathbf{W}$ in our dataset. This creates the necessity to *share parameters* across texts $\mathbf{W}$ to estimate the variance of $\mathbf{p}_t$ and hence $\text{H}(\mathbf{P}_t \mid \mathbf{W})$ on held-out data. By sharing parameters, we mean that we estimate $\text{H}(\mathbf{P}_t \mid \mathbf{W})$ for all $\mathbf{W}$ values via one function, which in our case is a language model (as explained in §2.2 Eq. (5)).

Why is parameter sharing suitable in our case? In language, unlike many other domains, distinct word sequences share inherent properties. For instance, the sentences *I love dogs* and *I love cats* hint at functional and semantic connections between *dogs* and *cats*. Thus, given prosodic values from the *dog*-example, we may be able to infer the prosodic feature variance of the word *cats*. By using pretrained neural language models—and hence their implicitly encoded linguistic knowledge—we share the neural network's parameters across linguistically related examples to make predictions about their prosodic features.

## 2.2 Estimation through cross-entropy

In this section, we explain how, given the data scarcity problem, we get estimates of the mutual information between prosody and text. Under the good mixed-pair assumption, we estimate the mutual information $\text{MI}(\mathbf{P}_t; \mathbf{W})$ as the difference between two cross-entropies $\text{H}_{\boldsymbol{\theta}}$ (McAllester and Stratos, 2020):

$$\text{MI}(\mathbf{P}_t; \mathbf{W}) = \text{H}(\mathbf{P}_t) - \text{H}(\mathbf{P}_t \mid \mathbf{W}) \qquad (3a)$$
$$\approx \text{H}_{\boldsymbol{\theta}}(\mathbf{P}_t) - \text{H}_{\boldsymbol{\theta}}(\mathbf{P}_t \mid \mathbf{W}) \qquad (3b)$$

which leaves us with the two differential cross-entropies in (3b) to estimate. These cross-entropies are defined as the expectation of $-\log p_{\boldsymbol{\theta}}(\mathbf{p}_t)$ with respect to the ground truth probability of the prosodic feature $p(\mathbf{p}_t)$ or, in the case of the conditional cross entropies $-\log p_{\boldsymbol{\theta}}(\mathbf{p}_t \mid \mathbf{w})$ and $p(\mathbf{p}_t, \mathbf{w})$.[6] In order to approximate these expectations, we use *resubstitution estimation* (Hall and Morton, 1993; Ahmad and Lin, 1976), using a set of held-out samples from our dataset.

---

[4] Note that $\text{MI}(\mathbf{P}_t; \mathbf{W}) = \text{MI}(\mathbf{W}; \mathbf{P}_t)$.

[5] The good mixed-pair assumption requires that $p(\mathbf{p}_t \mid \mathbf{w})$ be absolutely continuous with respect to $p(\mathbf{p}_t)$ for all $\mathbf{w} \in \Sigma^*$.

[6] Note that since the ground-truth probability $p$ is a probability density function, it can (unlike probability mass functions on discrete units) attain values larger than 1. This means that differential entropy can take on negative values and is not bounded from below. The mutual information, however, is still always non-negative.

Our cross-entropy estimates are thus:

$$\mathrm{H}_{\boldsymbol{\theta}}(\mathbf{P}_t) \approx \frac{1}{N} \sum_{n=1}^{N} \log \frac{1}{p_{\boldsymbol{\theta}}(\mathbf{p}_t{}^{(n)})} \tag{4a}$$

$$\mathrm{H}_{\boldsymbol{\theta}}(\mathbf{P}_t \mid \mathbf{W}) \approx \frac{1}{N} \sum_{n=1}^{N} \log \frac{1}{p_{\boldsymbol{\theta}}(\mathbf{p}_t{}^{(n)} \mid \mathbf{w}^{(n)})} \tag{4b}$$

where $N$ is the number of held-out samples. Importantly, the cross-entropy upper-bounds the entropy. It follows that the better our estimates $p_{\boldsymbol{\theta}}$ of $p$, the closer our cross-entropies will be to the actual entropies, and the better our estimation of the mutual information should be (Pimentel et al., 2020).

**Estimating $p_{\boldsymbol{\theta}}(\mathbf{p}_t)$.** We approximate the true distribution $p(\mathbf{p}_t)$ with a Gaussian kernel density estimate (Sheather, 2004). For details on this estimation method, we refer the reader to App. A.2.

**Estimating $p_{\boldsymbol{\theta}}(\mathbf{p}_t \mid \mathbf{w})$.** In order to approximate this conditional probability density function, we use a neural language model to *learn* the parameters of a predictive distribution, which is defined here as either a Gaussian or a Gamma distribution over the prosodic feature space $\mathbb{R}^d$:

$$\phi = \mathrm{LM}_{\boldsymbol{\theta}}(\mathbf{w}) \tag{5a}$$

$$p_{\boldsymbol{\theta}}(\mathbf{p}_t \mid \mathbf{w}) = \mathcal{Z}(\mathbf{p}_t; \phi) \tag{5b}$$

where we denote the predictive distribution with $\mathcal{Z}$ and its parameters with $\phi$. The neural network is itself noted as $\mathrm{LM}_{\boldsymbol{\theta}}$. As explained in §2.1, this allows us to share parameters across $\mathbf{w}$, where we only have one example for each $\mathbf{w}$ in our finite dataset. We note that $\mathrm{LM}_{\boldsymbol{\theta}}$ should learn to output similar values $\phi$ when input similar sentences $\mathbf{w}$, thus sharing parameters across sentences. However, since we use parametrized distributions $\mathcal{Z}$ to estimate $p_{\boldsymbol{\theta}}(\mathbf{p}_t \mid \mathbf{w})$, our estimator of the conditional entropy is not consistent (meaning that it might not converge to the true value as $N \to \infty$).

## 3 The role of context

While in §2 we deliberately kept the text-valued random variable $\mathbf{W}$ underspecified, we now introduce it more precisely to quantify the redundancy of prosody and different parts of the text. We measure the redundancy of prosody at a word $t$, which we write as $\mathbf{P}_t$, with three different text-valued random variables: a single word $\mathbf{W}_t$, all past words $\mathbf{W}_{\leftarrow}$, and all other words in a sentence $\mathbf{W}_{\leftrightarrow}$. We then estimate these measures using

uncontextualized word embeddings, as well as autoregressive and bidirectional language models. We discuss each of these setups, briefly, below.

### 3.1 Context types

**Current Word $\mathbf{W}_t$.** Given only the target word at index $t$ of a text, we estimate the mutual information between $\mathbf{W}_t$ and its word-level prosodic feature $\mathbf{P}_t$: $\mathrm{MI}(\mathbf{P}_t; \mathbf{W}_t)$. This lets us quantify how much information about the prosodic feature is encoded by the word type alone. To estimate the predictive distribution $p_{\boldsymbol{\theta}}(\mathbf{p}_t \mid \mathbf{w}_t)$—on top of the uncontextualized word embeddings—we train a multilayer perceptron that predicts the parameters $\phi$ of the predictive distribution. For more details, we refer to §4.3.1.

**Past context $\mathbf{W}_{\leftarrow}$.** Given the target word and its preceding words, we estimate the mutual information between the autoregressive context and the prosodic feature $\mathbf{P}_t$ of the target word: $\mathrm{MI}(\mathbf{P}_t; \mathbf{W}_{\leftarrow})$. We finetune an autoregressive language model (such as, e.g., GPT-2) to estimate $p_{\boldsymbol{\theta}}(\mathbf{p}_t \mid \mathbf{w}_{\leftarrow})$, which we describe in §4.3.2.

**Bidirectional context $\mathbf{W}_{\leftrightarrow}$.** Given the target word and the full context of the text sequence (i.e., the preceding as well as following context), we estimate the mutual information between the bidirectional context and the target word's prosodic feature $\mathbf{P}_t$: $\mathrm{MI}(\mathbf{P}_t; \mathbf{W}_{\leftrightarrow})$. We finetune a bidirectional language model (such as, e.g., BERT) to estimate $p_{\boldsymbol{\theta}}(\mathbf{p}_t \mid \mathbf{w}_{\leftrightarrow})$, which we describe in further detail in §4.3.2. Using the estimators above and the definition of conditional mutual information we can also infer the information shared exclusively between the future context $\mathbf{W}_{\rightarrow}$ and prosody, but not contained in the autoregressive context, by subtraction of two mutual informations:

$$\mathrm{MI}(\mathbf{P}_t; \mathbf{W}_{\rightarrow} \mid \mathbf{W}_{\leftarrow}) \tag{6}$$
$$= \mathrm{MI}(\mathbf{P}_t; \mathbf{W}_{\leftrightarrow}) - \mathrm{MI}(\mathbf{P}_t; \mathbf{W}_{\leftarrow})$$

## 4 Our prosody pipeline

In this section, we describe the data processing pipeline we use to extract the prosodic features energy, duration, pause, and pitch curve from word-level text-speech alignments.

### 4.1 Dataset

Our experiments use the LibriTTS corpus (Zen et al., 2019), created explicitly for text-to-speech

research. This corpus was developed from the audio and text materials present in the LibriSpeech corpus (Panayotov et al., 2015) and contains 585 hours of English speech data at a 24kHz sampling rate. The dataset includes recordings from 2,456 speakers and the corresponding transcripts. We use the data split provided by Zen et al. (2019) (see Table 2). Further, we only consider texts from LibriTTS that contain at least two words (since LibriTTS contains fragments like the book and chapter titles, e.g., *"Chapter IX"*).

In order to extract word-level prosodic features, we derive word-level alignments between audio and text using the Montreal Forced Aligner (McAuliffe et al., 2017).[7] In addition to the word-level alignment, we compute phoneme-level alignments to locate a word's stress syllable, which is necessary for extracting pitch contours (see §4.2).

## 4.2 Representations of prosodic features

Given the spoken utterances from the LibriTTS dataset, we need to extract the speakers' prosody. Prior work (Ye et al., 2023) has used the complete audio signal, extracting information about prosody only through contrasting identical words across different text-speech pairs. The full audio signal—due to the data processing inequality[8]—provides an upper bound on prosody-related information, but also contains information unrelated to prosody, such as phonetic features. In order to isolate suprasegmental prosodic elements specifically and get a better estimate of the mutual information between prosody and text, we instead choose to extract specific acoustic features considered relevant for prosody based on previous research. We concentrate on three basic acoustic properties: **Energy**, **duration**, and $f_0$. We also examine prosodic **prominence** as a composite signal of energy, duration, and $f_0$, which was suggested to capture functionally relevant supra-segmental stress patterns in English efficiently (Suni et al., 2017; Talman et al., 2019). We further examine the duration of a potential pause after a word, which has been discussed previously as an important cue for grammatical boundaries (Hawkins, 1971).

**Energy** Energy quantifies intensity variations in the speech signal , which are affected by sub-glottal pressure (i.e., the pressure of the air passing from the lungs to the vocal folds) as well as other articulatory-phonetic factors (for example, producing stop consonants, e.g., [k] or [p]). We quantify these intensity variations as *energy* and compute its values by first applying a bandpass filter[9] and then computing the logarithm of the amplitude of the filtered acoustic signal (see Suni et al. (2017)). We consolidate the word's energy into a scalar feature by averaging all sample points between the word boundaries. Specifically, we sum up the log amplitude values at each acoustic sampling point (we use 24kHz samples, see §4.1) and divide the sum by the number of sampling points to obtain a scalar measure.

**Duration and pause** We measure the duration of words and pauses separately. Word durations are computed from word-level text–speech alignment boundaries, and are normalized by a word's syllable count, obtained from the CELEX dataset (Baayen et al., 1995), to yield a scalar capturing the average syllable duration. In the case that a following pause exists, we measure the interval of silence between a word's offset and the next word's onset in our text–audio alignments. The pause duration is 0 for 89.4% of the words in our dataset.

**Pitch** Whereas other prosodic features can be represented as single scalar values without losing much information, it is well-established that the entire pitch contour contributes to the meaning (e.g., Pierrehumbert and Beckman, 2003). We apply the fundamental frequency ($f_0$) preprocessing methods detailed in Suni et al. (2017), which include $f_0$ extraction from raw waveforms, and outlier removal. Additionally, interpolation is applied at time points where the estimation algorithm fails due to silence or unvoiced phonemes leading to pitch-less sounds.

Rather than looking at the $f_0$ curve over the whole word, we focus on the area around the word's primary stressed syllable. This is motivated by the fact that English pitch accents anchor to the stressed syllable, and the same underlying pitch accent may be realized with phonologically irrelevant regions of the flat pitch in polysyllabic words (Pierrehumbert, 1980). We identify the stressed syllable using our phoneme level alignments and

---

[7]We used the English (US) Arpa acoustic model and dictionary (McAuliffe and Sonderegger, 2022).

[8]The data-processing inequality (Cover and Thomas, 2005, Chapter 2) states that for any function $f(\cdot)$, e.g., one that extracts prosody from audio ($f(\mathbf{A}_t) = \mathbf{P}_t$), we have: $\mathrm{MI}(\mathbf{A}_t; \mathbf{W}) \geq \mathrm{MI}(\mathbf{P}_t; \mathbf{W})$.

[9]The bandpass filter removes all frequencies smaller than 300Hz and larger than 5000Hz.

stress patterns from CELEX (Baayen et al., 1995) and extract up to 250ms of audio signal (less if the interval to word onset or offset is shorter, which is often the case) on either side of the stressed sylla­ble's midpoint.[10]

Instead of working with the raw $f_0$ curves, we translate them into a low-dimensional space. We first resample the $f_0$ curves to 100 sampling points and parameterize them with $k$ coefficients of a discrete cosine transform (DCT). For additional experiments, see App. B.4.

**Prominence** We use continuous word-level prominence values using the toolkit provided by Suni et al. (2017), combining energy (intensity), $f_0$ , and word duration into a single scalar promi­nence value per word. The authors perform a co­sine wavelet transform on the composite signal of energy, $f_0$, and duration and extract prominence as lines of maximum amplitude in the spectrum. Note that the processing of word duration, energy, and $f_0$ is distinct from the procedures introduced in the previous sections (see (Suni et al., 2017) for details) and we do not investigate the relative weighting of these features. Since the perception of prosodic prominence is relative to the prominence of the preceding words, we add another measure of the change in prominence relative to the average prominence of the three previous words. We refer to this as *relative prominence*.

### 4.3 Word representations

We choose to use orthographic representations for segmental information. Although there are other options, such as various phonetic transcriptions, this choice enables us to easily embed this informa­tion into a representational space using pretrained language models. We next present the implementa­tions for estimating the distributions conditioned on the three types of context discussed in §3, namely, distributions conditioned on word types, autore­gressive contexts and bidirectional contexts.

#### 4.3.1 Non-contextual estimators

We estimate $p_\theta(\mathbf{p}_t \mid \mathbf{w}_t)$ using fastText (Bo­janowski et al., 2017) and a multi-layer perceptron (MLP) to predict the parameters of the predictive distribution of a prosodic feature. Our MLPs

parameterize a predictive distribution over the prosodic feature space (e.g., a Gaussian or a Gamma distribution)[11] and are optimized via stochastic gradient descent using a cross-entropy loss. We choose our model's hyperparameters, i.e., learning rate, $L_2$-regularization, dropout probability, number of layers, and hidden units, with 50 iterations of random search.

#### 4.3.2 Contextual estimators

For the distributions conditioned on multiple words, we use a fine-tuning setup to approximate $p_\theta(\mathbf{p}_t \mid \mathbf{w}_\leftarrow)$ and $p_\theta(\mathbf{p}_t \mid \mathbf{w}_\leftrightarrow)$. In our setup, autoregres­sive models $p_\theta(\mathbf{p}_t \mid \mathbf{w}_\leftarrow)$ can be thought of as useful for modeling incremental language compre­hension, where the model's predictions at a target word are only affected by the preceding context. We use GPT-2 (Radford et al., 2019) models rang­ing from 117M to 1.5B parameters for our autore­gressive models $\mathrm{LM}_\theta(\mathbf{w}_\leftarrow)$. We fine-tune these LMs to parameterize a predictive distribution over the prosodic feature space using a cross-entropy loss. After fine-tuning several models in this way, we select the best one based on their cross-entropy loss on a validation set. We use the pre-defined Lib­riTTS data split for training, validation and testing; see App. B.1 for details.

Bidirectional models are trained to predict prosody given information about both their left and right contexts, $p_\theta(\mathbf{p}_t \mid \mathbf{w}_\leftrightarrow)$. Comparing their performance to the autoregressive models can therefore teach us specifically about the gain in mutual information between prosody and text con­tributed by the words following the target word. This analysis allows us to quantify the extent to which prosody, itself, can be used in prediction dur­ing language processing, by revealing how much information it contains about upcoming material. We consider bidirectional LMs from the BERT-, RoBERTa- and DeBERTa families (Devlin et al., 2019; Liu et al., 2019; He et al., 2021) ranging from 110M to 1.5B parameters to implement $\mathrm{LM}_\theta(\mathbf{w}_\leftrightarrow)$. As with autoregressive models, we fine-tune the pretrained bidirectional LLMs to estimate the pa­rameters of a parameterized distribution using a

---

[10]One potential shortcoming of this procedure is that it will zero in on pitch events around the stressed syllable. In English, this will capture stress patterns but will potentially miss bound­ary tones occurring primarily at the end of prosodic phrases. We leave the investigation of boundary tones to future work.

[11]Note that for the prosodic features introduced in §4.2 we distinguish between two types. Prominence, energy, duration, and pause can be represented by a scalar value $\mathbf{p}_t \in \mathbb{R}$. While energy and duration are parameterized by a Gaussian, we chose to parameterize prominence and pauses by a Gamma distribution due to their heavy tails. The pitch curves are represented by 8 coefficients of the discrete cosine transform of the curve $\mathbf{p}_t \in \mathbb{R}^8$ and is parameterized by a Gaussian.

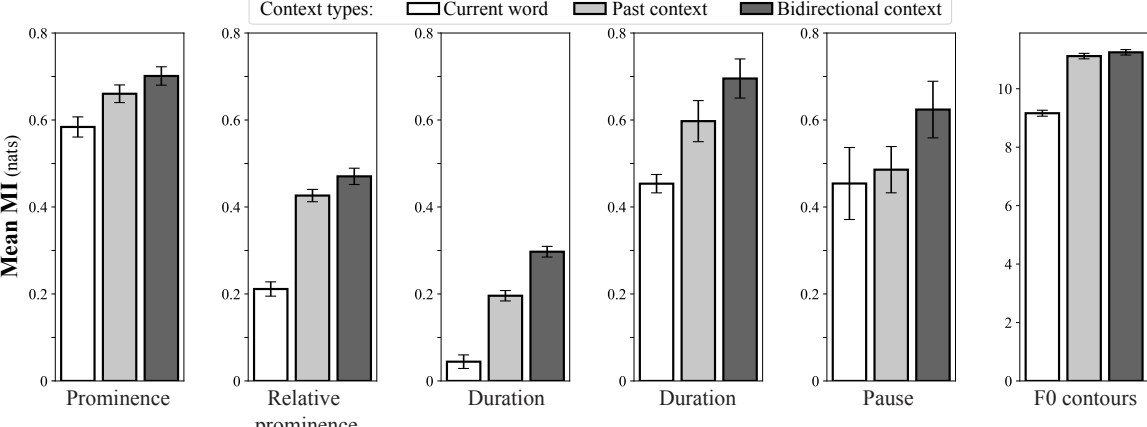

Figure 2: Estimated mutual information $\mathrm{MI}(\mathbf{P}_t; \mathbf{W}) = \mathrm{H}(\mathbf{P}_t) - \mathrm{H}(\mathbf{P}_t \mid \mathbf{W})$ for all models and features with mean and standard deviation from 5 runs of cross-validation. We fix the scale for the prosodic features prominence, relative prominence, energy, duration, and pause and report the $f_0$ (pitch) results on a separate axis due to its higher mutual information. Note that the differential entropy of $f_0$ curves (see Table 1) is much higher than the differential entropy of the other prosodic features. We report the mutual information in nats.

cross-entropy loss. As above, we select the best model based on their cross-entropy loss.

## 5 Experimental results

### 5.1 Correlates of prosodic prominence

Prior work (Malisz et al., 2018; Pimentel et al., 2021) indicates a correlation between word duration and surprisal. Prosodic prominence, as defined by Suni et al. (2017), is a composite measure of word duration, energy, and $f_0$. We perform a validation on our prosody pipeline to verify that word-level prosodic prominence correlates with surprisal. Using the next-word predictive distribution over the vocabulary of an autoregressive language model (GPT-2), we find a weak, positive correlation between prosodic prominence and surprisal, as well as between prosodic prominence and token entropy; see App. B.7.

### 5.2 Main results

Our primary results measuring the mutual information can be seen in Figure 2, with the different prosodic features in the different facets. First, when comparing the different context types, our results demonstrate a robust pattern across prosodic features: All of the prosodic feature–model pairs result in positive mutual information, indicating the presence of redundancy between prosody and text. Even the non-contextual estimator already produces positive mutual information with all prosodic features, indicating that prosodic cues can be pre-

dicted to some extent by the word type alone. Importantly, contextual estimators have higher mutual information values compared to the non-contextual estimators for all prosodic features, indicating that the surrounding words improve prosody predictions compared to just the individual word type. Furthermore, bidirectional estimators show higher mutual information values compared to autoregressive ones for all features apart from the pitch curves, which only show a very small increase. Below, we discuss each of the prosodic features in turn, moving from left to right across the facets.

With respect to absolute prominence, we find the least difference between the non-contextual estimator and both the contextual ones. This indicates that much of the information about absolute prominence is predictable from word type alone. This is not true, however, when we look at relative prominence, where the bidirectional estimator reveals more than 2.3 times as much information (0.48 nats) than the non-contextual estimator, fastText. We see overall lower mutual information for energy than for any of the other prosodic features, suggesting that energy is less tightly connected to segmental information. Like with relative prominence, we observe much higher estimated mutual information for the contextual estimators, and, interestingly, here, we observe one of the largest differences between our bidirectional and autoregressive estimators (with 66% more information obtained in the latter). Results for duration and pauses are similar. Like prominence, the non-contextual estimator

| Feature | Diff. Entropy $H(\mathbf{P}_t)$ | Current Word $H(\mathbf{P}_t \mid W_t)$ | Past Context $H(\mathbf{P}_t \mid W_\leftarrow)$ | Bidirectional Context $H(\mathbf{P}_t \mid W_\leftrightarrow)$ |
|---|---|---|---|---|
| Prominence | $0.536 \pm 0.012$ | $-0.048 \pm 0.005$ | $-0.124 \pm 0.002$ | $-0.165 \pm 0.003$ |
| Prominence relative | $1.355 \pm 0.011$ | $1.145 \pm 0.005$ | $0.924 \pm 0.003$ | $0.878 \pm 0.008$ |
| Energy | $0.814 \pm 0.008$ | $0.749 \pm 0.001$ | $0.618 \pm 0.004$ | $0.516 \pm 0.004$ |
| Duration (per syl.) | $-0.920 \pm 0.009$ | $-1.416 \pm 0.012$ | $-1.506 \pm 0.038$ | $-1.657 \pm 0.036$ |
| Pause | $-5.189 \pm 0.050$ | $-5.537 \pm 0.033$ | $-5.676 \pm 0.003$ | $-5.813 \pm 0.015$ |
| $f_0$ curves | $11.619 \pm 0.093$ | $2.477 \pm 0.009$ | $0.719 \pm 0.001$ | $0.662 \pm 0.001$ |

Table 1: The estimated differential entropy of the $H(\mathbf{P}_t)$ and the conditional entropy $H(\mathbf{P}_t \mid W)$ for all prosodic features and word representations. For differential entropy, standard deviations are from bootstraps over 20 folds of data. We choose the **best autoregressive and bidirectional model** based on the validation loss and report the conditional entropy of these models on the held-out test data. For the non-contextual estimator based on fastText, MLP hyperparameters are optimized via 50 iterations of random grid search, and we report the test set conditional entropy of the best validation model.

achieves relatively high mutual information for both, indicating that they can be partially predicted from word-level factors. For both, we see gains for our autoregressive estimators over non-contextual ones and for the bidirectional estimators over the autoregressive ones. Likely, this is due to increased pauses (if there is a pause, i.e. duration of the pause after a word is greater than 0) and increased duration at the end of prosodic phrases, which can be well-predicted by the bidirectional estimator, which has access to information about following words as well as upcoming punctuation.

Finally, for pitch, we find high values of mutual information for all estimators. This is due to the fact that our representation of pitch (i.e., 8 DCT coefficients) contains more information than the other features, as obtained by our estimates of (un-conditional) differential entropy (Table 1). For the pitch curves, bidirectional context does not contain much more information than autoregressive one, suggesting that the pitch events of a particular word are mostly related to the preceding context and less so to the following context. Representing the pitch curves with more DCT coefficients increased the mutual information with the bidirectional context (Table 3), but we did not run the autoregressive estimators for the different number of DCT parameters to check if the non-improvement holds for bidirectional context holds as well.

## 6 Discussion

We studied the interaction between speech prosody and linguistic meaning conveyed by the words themselves via computing the mutual information (MI) between prosodic features extracted from a

corpus of spoken English audiobooks and its associated text. We found that, consistently across a variety of prosodic features, there exists non-trivial MI between the prosody of a given word and the word itself. For all prosodic features, there is an increase of the MI between prosody and text when considering the preceding context as well. Furthermore, for most prosodic features, considering the words following the target word further increases the MI with the prosody. Our results quantitatively reveal the partial redundancy between prosodic features and linguistic content extracted from text alone.

**Our large-scale approach to prosody.** Much previous research on prosody concentrated on characterizing specific prosodic features and identifying their contributions to specific linguistic functions. In contrast to these previous studies, our methods facilitated a large-scale systematic examination of the statistical properties of various prosodic features and their MI with abstract text representations extracted using language models. As such, our approach lacks some interpretability relative to previous studies, as we do not know what are the specific linguistic functions associated with the information that we found to be redundant with certain prosodic features. The latter could be further explored in future work by comparing text where prosody predictions were especially high or low. However, a big advantage of our approach compared to previous studies is the quantitative large-scale examination of a spoken linguistic corpus, thus potentially revealing more mutual information between prosody and linguistic information than was previously possible to show, because the models may extract abstract linguistic representations that were

not specifically targeted by previous studies.

**The role of context.** The positive MI of the non-contextualized estimators indicates that there is already information in word identity about prosodic features. This indicated that some words tend to have different prototypical prosodic values than other words. This is consistent with past studies suggesting that the cognitive representation of words interacts with their typical prosody (Seyfarth, 2014; Tang and Shaw, 2021). Comparing the MI of prosody with past vs. bidirectional context can give insight into real-time language production and comprehension. For production, the gain in MI for both context types indicates that language production systems (whether in humans or machines) would be well-served by having a representation of the full utterance in order to effectively plan and execute the prosodic features of their speech. As far as online comprehension, the results for bidirectional contexts suggest two things: First, it is not just the case that linguistic content can be used to predict prosody, but that prosodic information could be used to predict future textual information, insofar as there is redundant information between the two. Second, they suggest that during comprehension, the meaning of a prosodic event may only be fully perceived at the end of the utterance.

**Differences between prosodic features.** Although our results suggest that most prosodic features gain a significant amount of information from past context as well as future context (Figure 2), there are some interesting details and exceptions. F0 contours as well as relative prominence gain most of their information from past and much less from future context. In contrast, pauses gain most of their information with text from future context and almost none from past context. The relatively larger importance of future context for pauses we interpret as confirming previous work on the relationship between pauses and syntactic boundaries (Hawkins, 1971), where the following words can help reveal whether a current word marks the end of a syntactic unit. One potential confounding factor for all these points, however, is the effect of punctuation marks, which are considered in the bidirectional but not the autoregressive estimators. In future research, we plan to disentangle these two to better understand the effect of the following words factoring out punctuation marks, which are not present in spoken communication.

It is important to mention that the order of magnitude difference in the mutual information scales between the f0 contours and the rest of the prosodic features (Figure 2) is due to their representation as multidimensional features. We represented the full f0 curve with a dimensionality reduction achieved by a DCT decomposition and found that 8 DCT components were favorable in terms of MI maximization. Future work can use our methodology to further explore the prosodic functions of f0 curves.

**Future work.** Exploring the interaction between linguistic communication channels holds a deep significance in understanding the multidimensional distribution of meaning in language, which extends beyond prosody to include facets like gestures and facial expressions. The implications of this understanding are profound. For example, the excess information contained in prosody could set a theoretical performance ceiling for text-to-speech systems, which are tasked with generating speech prosody from text alone. The integration of prosodic information could aid neural language model training, which, despite excelling in machine translation (Vaswani et al., 2017) and language generation (Brown et al., 2020), predominantly rely on text data. As prosody is fundamental to spoken language, incorporating it may address the current endeavor to boost language model performance using developmentally realistic data volumes (Warstadt et al., 2023; Wolf et al., 2023).

## 7    Conclusion

We presented a systematic large-scale analysis of the mutual information between a variety of suprasegmental prosodic features and linguistic information extracted from text alone, using the most powerful openly available language models. The theoretical framework we developed to estimate mutual information between these two communication channels, as well as our openly available data processing implementation, can be used to investigate interactions between other communication channels. These tools enabled us to quantitatively demonstrate that information conveyed by speech prosody is partly redundant with information that is contained in the words alone. Our results demonstrate a relationship between the prosody of a single word and the word's identity, as well as with both past and future words.

## Limitations

Our paper has two main empirical limitations. First, as the quantities of mutual information we are interested in are not analytically computable, we rely on estimators. We do not, however, have a good sense of the errors associated with these estimators. Further, as our quantities rely on differential entropy, which is unbounded below, our cross-entropy upper bound estimates (as in, e.g., Eq. (12)) could be arbitrarily loose. Increasing the size of the dataset and using the strongest available models is important in order to improve our estimation of the MI between prosody and text. Our work, however, offers a general methodology which future work can apply using potentially improved models—either to study prosody itself, or other linguistic features.

Second, our work examines LibriTTS as its main source of data. This dataset is itself composed of audiobooks made available by the Librispeech dataset (Panayotov et al., 2015); and is composed exclusively of English sentences. As this dataset only covers audiobooks, the generalizability of our results might be limited given that prosody is highly dependent on speaking styles. Furthermore, our work obviously applies just for English, whereas the role of prosody in conveying meaning is highly dependent on the specific language. Future work will apply our approach using more datasets to add other domains of speech and languages. Important questions could be answered by applying our approach to different speech types and languages: For example, this could be a quantitative tool to systematically examine the universality of prosody across languages, or to study the role of prosody in language learning using recordings of child-directed speech.

## Ethics Statement

Our research primarily involves the analysis of linguistic data and models and does not directly conflict with ethical norms or privacy concerns. We have used publicly available datasets, which do not contain any personally identifiable information or any sensitive content. However, we acknowledge that research in language modeling and prosody has potential applications that need to be approached with ethical caution. One such application is text-to-speech synthesis (TTS). Advanced TTS systems, especially when integrated with language models capable of generating human-like speech, could be used for imitating voices. Considering these poten-tial applications and their ethical implications, we encourage the academic and industrial community to approach the deployment of such technologies with caution and to consider their societal impact.

## Contribution Statement

LW, TIR, EGW and AW conceived of the ideas presented in this work. LW, TP and RC developed the mathematical portions of the estimation methods. LW implemented the methods and carried out the experiments. EF contributed to early discussions. LW, TP, EGW, AW and TIR wrote the first draft of the manuscript. All authors edited the manuscript and reviewed the work. EF and RC provided funding for the present work.

## Acknowledgements

EGW would like to acknowledge support from an ETH Postdoctoral Fellowship. TIR would like to thank Noga Zaslavski for early discussions. TIR is supported by the Poitras Center for Psychiatric Disorders Research at the McGovern Institute, MIT. TIR and EF were supported by the MIT Research Support Committee. All authors would like to thank Jennifer Cole for her helpful insights on pitch curve representation, and Eleanor Chodroff for thorough feedback on the phonetics portion of the manuscript. Any remaining errors in the work are surely our own.

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

## A   Mixed-pair mutual information estimation

### A.1   Challenges in mixed-pair mutual information estimation

We now expand on the problem of estimating the mutual information between discrete–continuous pairs of random variables. Many common mutual information estimators rely on the decomposition in Eq. (2) (Sricharan et al., 2013; Gao et al., 2016; Han et al., 2015), and might thus not be applicable to our mixed setting. Estimators that rely on a decomposition such as in Eq. (2) adhere to the 3H-principle, where the mutual information is decomposed into three entropies: $\mathrm{MI}(\mathbf{P}_t, \mathbf{W}) = \mathrm{H}(\mathbf{P}_t) + \mathrm{H}(\mathbf{W}) - \mathrm{H}(\mathbf{P}_t, \mathbf{W})$. Other popular approaches for mutual information estimation include non-parametric kernel density estimators (Fraser and Swinney, 1986; Moon et al., 1995; Darbellay and Vajda, 1999; Suzuki et al., 2008; Kraskov et al., 2004). Many of these, however, were not specifically developed for continuous–discrete distributions.

Due to this issue, a few estimators have recently been proposed specifically for a continuous–discrete case in which Eq. (2) might not hold. For instance, Gao et al. (2017) propose a $k$-nearest neighbor ($k$-NN) based estimator for the mutual information, which they prove to be consistent for continuous–discrete distributions, meaning that it converges to the true mutual information given enough data. Alternatively, Beknazaryan et al. (2019) give a simple sufficient condition for the decomposition in Eq. (2) to hold, which they term the **good mixed-pair assumption.** The good mixed-pair assumption requires that $p(\mathbf{p}_t \mid \mathbf{w})$ be absolutely continuous with respect to $p(\mathbf{p}_t)$ for all $\mathbf{w} \in \Sigma^*$. Under this assumption, we *can* decompose the mutual information as:

$$\mathrm{MI}(\mathbf{P}_t; \mathbf{W}) = \sum_{\mathbf{w} \in \Sigma^*} \int_{\mathbb{R}^d} p(\mathbf{p}_t, \mathbf{w}) \log \frac{p(\mathbf{p}_t \mid \mathbf{w})}{p(\mathbf{p}_t)} \mathrm{d}\mathbf{p}_t$$
$$= \mathrm{H}(\mathbf{P}_t) - \sum_{\mathbf{w} \in \Sigma^*} p(\mathbf{w}) \, \mathrm{H}(\mathbf{P}_t \mid \mathbf{W} = \mathbf{w}) \tag{7}$$

Beknazaryan et al. then propose to use plug-in kernel density estimators to estimate $\mathrm{H}(\mathbf{P}_t)$ and each $\mathrm{H}(\mathbf{P}_t \mid \mathbf{W} = \mathbf{w})$ individually and prove such an estimator is consistent. As we explained in §2.1, this estimator may not apply to our case because of the data scarcity we encounter in our dataset.

### A.2   Estimating the mutual information with cross-entropies

**Estimating the unconditional entropy**   Differential entropy is an expectation of the negative log probability density. Therefore, we can estimate it by sampling from the true distribution and forming an empirical average of the negative log probability density. This empirical average will converge to the true differential entropy by the law of large numbers. We use Gaussian kernel density estimation (GKDE) (Sheather, 2004) to estimate the probability density function $p(\mathbf{p}_t)$ of a prosodic feature. Specifically, given a training set $\mathcal{D}_{\mathrm{trn}} = \{(\mathbf{p}_t^{(n)}, \mathrm{w}^{(n)})\}_{n=1}^{N_{\mathrm{trn}}} \sim p(\mathbf{p}_t, \mathbf{w})$ we obtain the kernel density estimation at a point $\mathbf{p}_t$ as:

$$p_{\boldsymbol{\theta}}(\mathbf{p}_t) = \frac{1}{N_{\mathrm{trn}} \, \mathrm{h}} \sum_{n=1}^{N_{\mathrm{trn}}} \mathrm{K}\left(\mathbf{p}_t, \mathbf{p}_t^{(n)}, \Sigma_{\mathcal{D}_{\mathrm{trn}}}, \mathrm{h}\right) \tag{8}$$

where $\mathrm{h} > 0$ is a smoothing parameter called the *bandwidth* and $\mathrm{K}$ is a Gaussian kernel.[12] GKDE is a non-parametric estimation method, i.e., it does not assume any underlying statistical distribution for the input data. This is particularly useful when the nature of the data is unknown or not well modeled by standard distributions.

We choose the hyperparameter configuration that achieves the highest likelihood on a held-out set of data points. After calculating the density estimate $p_{\boldsymbol{\theta}}$, we sample N points $\mathbf{p}_t^{(n)} \sim p(\mathbf{p}_t)$ from our test

---

[12]To relate this to the Gaussian kernel function, consider $\mathbf{p}_t$ and $\mathbf{p}_t^{(n)}$ to be $d$-dimensional vectors, and $\Sigma$ to be the $d \times d$ covariance matrix. The default choice of most existing libraries (e.g., SciPy Gaussian kernel density estimation) is to scale the covariance matrix of the dataset with the bandwidth h. The Gaussian kernel is then given as: $\mathrm{K}(\mathbf{p}_t, \mathbf{p}_t^{(n)}, \Sigma, \mathrm{h})$
$= \frac{1}{\sqrt{(2\pi)^d |\mathrm{h}\Sigma|}} \exp\left(-\frac{1}{2}(\mathbf{p}_t - \mathbf{p}_t^{(n)})^T (\mathrm{h}\Sigma)^{-1}(\mathbf{p}_t - \mathbf{p}_t^{(n)})\right)$

dataset $\mathcal{D}_{\text{tst}}$ (where $\mathbf{p}_t^{(n)}$ are the samples from the true distribution $p(\mathbf{p}_t)$), and approximate the differential entropy of the feature with Monte-Carlo sampling:

$$\mathrm{H}(\mathbf{P}_t) = -\int_{\mathbb{R}^d} p(\mathbf{p}_t) \log p(\mathbf{p}_t) \mathrm{d}\mathbf{p}_t \tag{9}$$

$$\leq -\int_{\mathbb{R}^d} p(\mathbf{p}_t) \log p_{\boldsymbol{\theta}}(\mathbf{p}_t) \mathrm{d}\mathbf{p}_t \approx -\frac{1}{N} \sum_{n=1}^{N} \log p_{\boldsymbol{\theta}}(\mathbf{p}_t^{(n)})$$

Note the last $\approx$ becomes exact as $N \to \infty$.

**Estimating the conditional entropy**   In order to approximate the conditional probability density function $p(\mathbf{p}_t \mid \mathbf{w})$, we use a neural language model to *learn* the parameters of a predictive distribution conditioned on some text $\mathbf{w}$. This allows us to share parameters across $\mathbf{w}$, where we only have one examples for each $\mathbf{w}$ in our finite dataset. This predictive distribution $p_{\boldsymbol{\theta}}(\mathbf{p}_t \mid \mathbf{w})$ is defined here as either a Gaussian or a Gamma distribution over the prosodic feature space $\mathbb{R}^d$

$$\boldsymbol{\phi} = \mathrm{LM}_{\boldsymbol{\theta}}(\mathbf{w}) \tag{10}$$
$$p_{\boldsymbol{\theta}}(\mathbf{p}_t \mid \mathbf{w}) = \mathcal{Z}(\mathbf{p}_t; \boldsymbol{\phi}) \tag{11}$$

where we denote the predictive distribution with $\mathcal{Z}$ and its parameters with $\boldsymbol{\phi}$. The neural network is itself noted as $\mathrm{LM}_{\boldsymbol{\theta}}$. In order to do so, we can choose any model that takes $\mathbf{w}$ as inputs and predicts the parameters of the predictive distribution. For each prosodic feature, we thus choose a predictive distribution based on model fit (see App. B.3). As we estimate an upper bound on the entropy, the better our model the tighter our entropy estimate will be (Pimentel et al., 2020). We note that $\mathrm{LM}_{\boldsymbol{\theta}}$ should learn to output similar values $\boldsymbol{\phi}$ when input similar sentences $\mathbf{w}$, thus sharing parameters across sentences.

We can use the predictive distribution to approximate the conditional entropy $\mathrm{H}(\mathbf{P}_t \mid \mathbf{W})$ by evaluating samples from the dataset. Therefore, we draw $N$ samples from a dataset, assumed to be distributed $(\mathbf{p}_t^{(n)}, \mathrm{w}^{(n)}) \sim p(\mathbf{p}_t, \mathbf{w})$, which allows us to approximate:

$$\mathrm{H}(\mathbf{P}_t \mid \mathbf{W}) = -\sum_{\mathbf{w} \in \Sigma^*} \int_{\mathbb{R}^d} p(\mathbf{p}_t, \mathbf{w}) \log p(\mathbf{p}_t \mid \mathbf{w}) \mathrm{d}\mathbf{p}_t \tag{12}$$

$$\leq -\sum_{\mathbf{w} \in \Sigma^*} \int_{\mathbb{R}^d} p(\mathbf{p}_t, \mathbf{w}) \log p_{\boldsymbol{\theta}}(\mathbf{p}_t \mid \mathbf{w}) \mathrm{d}\mathbf{p}_t \tag{13}$$

$$\approx -\frac{1}{N} \sum_{n=1}^{N} \log p_{\boldsymbol{\theta}}(\mathbf{p}_t^{(n)} \mid \mathrm{w}^{(n)}) \tag{14}$$

where, again, the last $\approx$ is exact as $N \to \infty$.

### A.3   Validation study

In order to understand how our method behaves under the data scarcity that we encounter in practice, we benchmark it against an established, kernel smoothing-based method.[13] We find (see Figure 3) that our estimator consistently underestimates the mutual information between prosody and text, which is consistent with our estimation being a lower bound for the mutual information. Therefore, the present study is more prone to Type II errors (not finding an effect although there is one) than to Type I errors (incorrectly finding an effect that does not exist), with the latter generally considered to be more severe.

## B   Experiment supplementary

### B.1   LibriTTS dataset

Details about the dataset can be found in Table 2.

---

[13]We use the mpmi R package which uses a kernel smoothing approach to calculate mutual information for comparisons between all types of random variables including continuous and discrete.

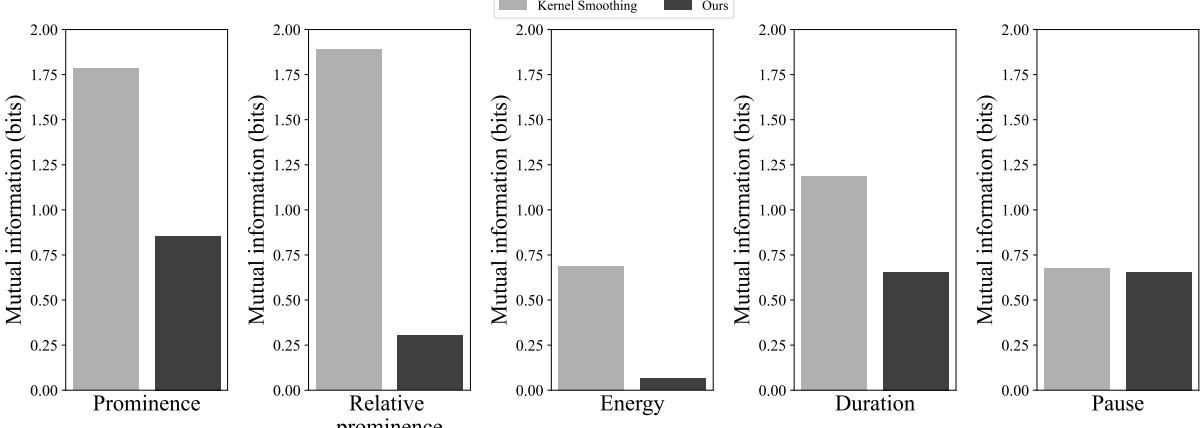

Figure 3: Comparison of Mutual Information Estimates for Prosodic Features: Kernel Smoothing Estimate vs. Ours. Notably, our method consistently underestimates the MI across all features. Note that we could not use the mpmi R package to estimate the mutual information between $f_0$ and text.

| Split | # Samples | # Words |
|-------|-----------|---------|
| Train | 116.2K    | 2378K   |
| Dev   | 5.7K      | 114K    |
| Test  | 4.8K      | 103K    |

Table 2: The number of samples and words in the LibriTTS dataset. We use the *train-clean-360* training split.

## B.2 Details on the prediction task

In order to compute the conditional entropies by fine-tuning large language models (LLMs), we treat the problem of predicting prosodic features as a token tagging task. We add an additional tagging head on the LM embeddings in order to predict the parameters of the predictive distribution. Since — depending on the tokenizer — a word might be represented by the tokens of the subwords that constitute it; we predict the prosodic feature of a word for all its tokens. The objective function is only applied to the last token of a word. We fine-tune the LMs by minimizing the negative log-likelihood of the target values under the predictive distribution.

## B.3 Visualizations of kernel density estimates

Figure 4 shows the estimated distributions of the two prominence features (absolute and relative). Figure 5 shows the estimated distributions for the features energy, $f_0$ (pitch), duration, and pause.

## B.4 Pitch curve parameterization

We provide additional experiments with alternative parameterizations of the pitch curve. We use the best model we found for this feature (RoBERTA-large[14]) and fine-tune the model with the same hyperparameter configuration for all pitch curve parameterizations. The estimated entropies for 2-, 4-, 8-, and 16 DCT coefficient parameterizations can be found in Table 3. We observe that even with an increased granularity of the curve parameterizations, the fine-tuned language model is to extract an increased amount of mutual information.

## B.5 Comment on grouped results

We find that no single model of our autoregressive and bidirectional groups performs best within a group across all features. The second-largest model, GPT2-large, performs best across most features for the

---

[14] https://huggingface.co/roberta-large

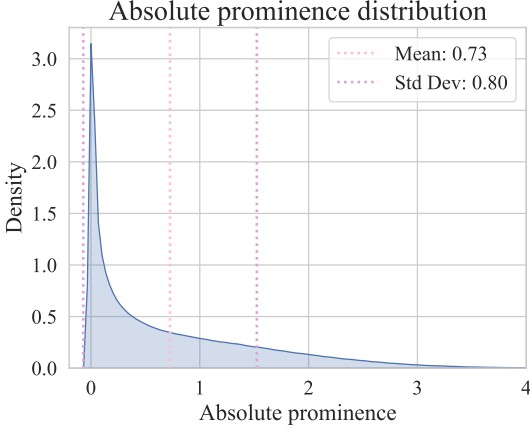

(a) Kernel density estimation of absolute prominence values of words.

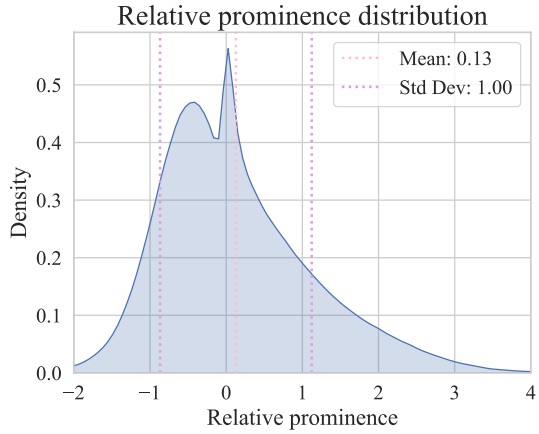

(b) Kernel density estimation of the prominence values of words relative to the three previous words in the text.

Figure 4: Estimated probability density functions of all prosodic prominence.

| # DCT Parameters | $\mathrm{H}(\mathbf{P}_t)$ | $\mathrm{H}(\mathbf{P}_t \mid \mathbf{W})$ | MI |
|:---:|:---:|:---:|:---:|
| 2 | 6.120 | 1.424 | 4.696 |
| 4 | 9.312 | 2.303 | 7.009 |
| 8 | 11.619 | 2.936 | 8.683 |
| 16 | 9.048 | 2.363 | 6.685 |

Table 3: Differential entropy $\mathrm{H}(\mathbf{P}_t)$, conditional differential entropy $\mathrm{H}(\mathbf{P}_t \mid \mathbf{W})$, and mutual information MI of alternative pitch curve parameterizations. We train the best model found (RoBERTa) with the same hyperparameter configuration on all parameterizations.

autoregressive language models. BERT and RoBERTa perform best across most prosodic features for the bidirectional models. Future work will include a larger dataset which allows for better fine-tuning of large models (e.g., Llama Touvron et al. (2023)).

## B.6 Baselining the uncertainty coefficient

In order to put the estimated entropies into context, we provide a **lower bound** on the conditional entropy by providing an entropy $\mathrm{H}_{\min}(\mathbf{P}_t)$ estimate that captures the uncertainty in the measurements. Precisely, $\mathrm{H}_{\min}$ tells us about the amount of information assuming a certain measurement error as a lower bound of precision. Assuming a conservative three-digit precision and uniform distribution over the least significant digit of the prosodic feature values leads to a lower bound of $\mathrm{H}_{\min}(\mathbf{P}_t) = -6.907$ for prominence, energy,

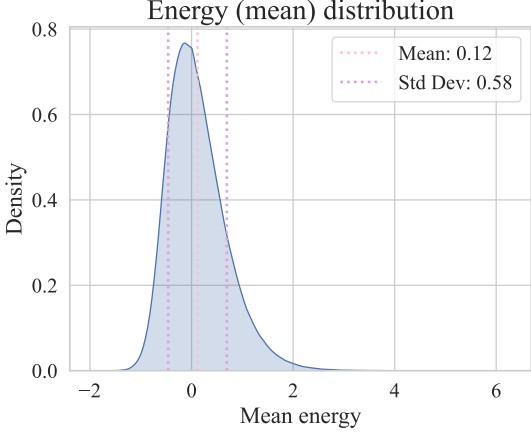

(a) Mean Energy per Word. The energy signal is normalized before computing the mean energy of each word.

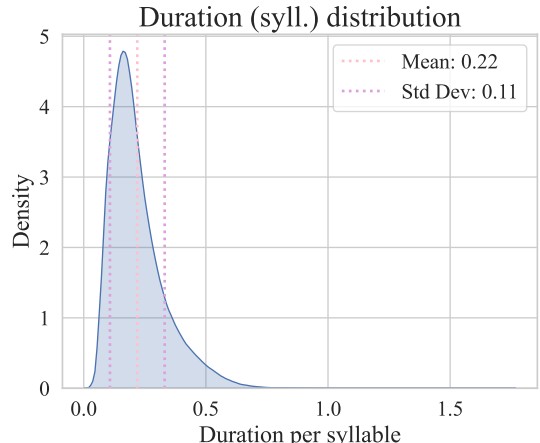

(b) Word duration in seconds per syllable.

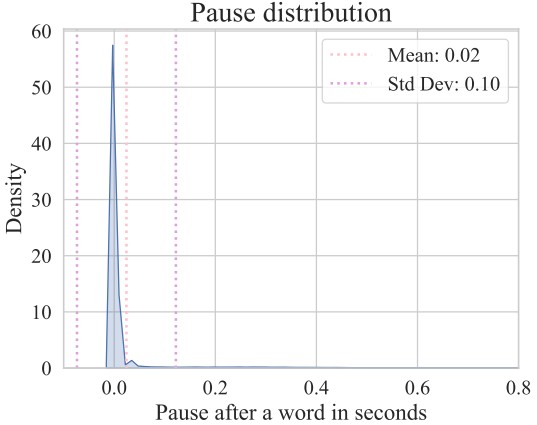

(c) Pause duration after a word in seconds.

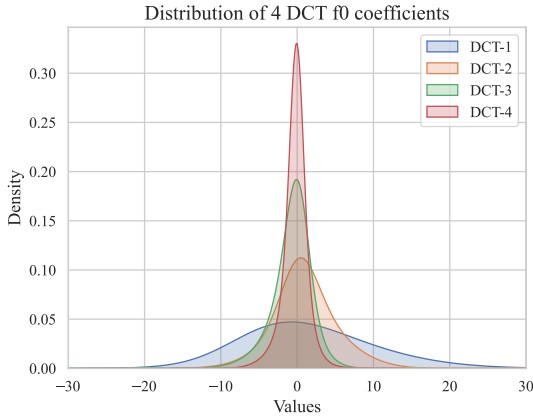

(d) The distributions over the four DCT coefficients parameterizing the pitch curve around the stress syllable.

Figure 5: Estimated probability density functions of prosodic features.

and pitch features.

Using this lower bound, we now introduce a *baselined* version of the widely known **uncertainty coefficient**[15]. The uncertainty coefficient tells us: given $\mathbf{W}$, what fraction of the information of $\mathbf{P}_t$ can we predict? In this case, we can think of $\mathbf{P}_t$ as containing the total information and of $\mathbf{W}$ as allowing one to predict part of such information. Since the coefficient makes use of an absolute lower bound, we obtain rather small values but are able to evaluate the different models on a meaningful range. A visualization of the uncertainty coefficients can be found in Figure 6.

$$\mathrm{U}(\mathbf{P}_t \mid \mathbf{W}) = \frac{\mathrm{MI}(\mathbf{P}_t; \mathbf{W})}{\mathrm{H}(\mathbf{P}_t) - \mathrm{H}_{\min}(\mathbf{P}_t)} \tag{15}$$

### B.7 Correlation of prominence and surprisal

Prominence is a composite feature of word energy, duration and pitch (Suni et al., 2017; Talman et al., 2019). We extend prior work on surprisal and duration (Seyfarth, 2014; Tang and Shaw, 2021; Pimentel et al., 2021) by concentrating on an analysis of word-level prominence features and computing the surprisal using powerful transformer architectures, modeling longer contexts than previous studies (Tang and Shaw, 2021). We find a weak, but positive correlation between word prominence and surprisal as well as word prominence and entropy (see Figure 7). We report the entropy of the distribution over the vocabulary given by the language model head of an autoregressive language model (GPT-2).

---

[15]The uncertainty coefficient is defined as the fraction between mutual information and entropy: $\mathrm{U}(\mathbf{P}_t \mid \mathbf{W}) = \frac{\mathrm{MI}(\mathbf{P}_t; \mathbf{W})}{\mathrm{H}(\mathbf{P}_t)}$.

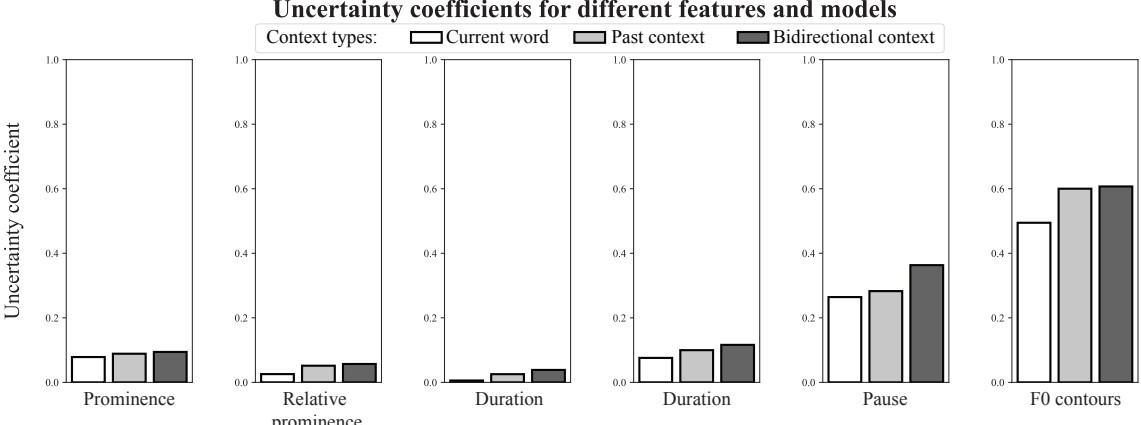

Figure 6: Estimated baselined uncertainty coefficient (UC) for all models and features.

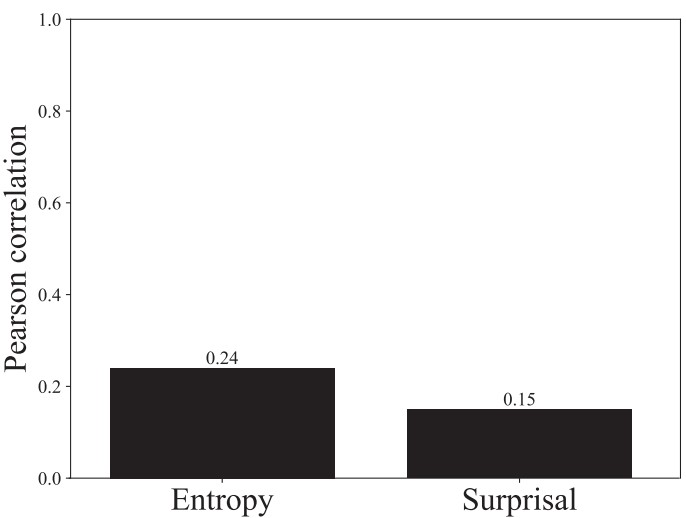

Figure 7: We show the correlation between the prominence of a token and its surprisal (negative log probability) as well as the entropy over the vocabulary distribution of the preceding token.

## B.8 Model inference

Our finetuned models can be used to perform inference to predict prosodic features directly from textual input. Notably, because our models parameterize an entire distribution, they also provide valuable insights into the probability density associated with each prediction. This enables a comprehensive understanding of the variability and confidence associated with each prosodic feature prediction. Example predictions (predicting the mean of the predictive distribution) of absolute prominence using the BERT model can be seen in Figure 8. We supply examples prediction for all features and models on GitHub.

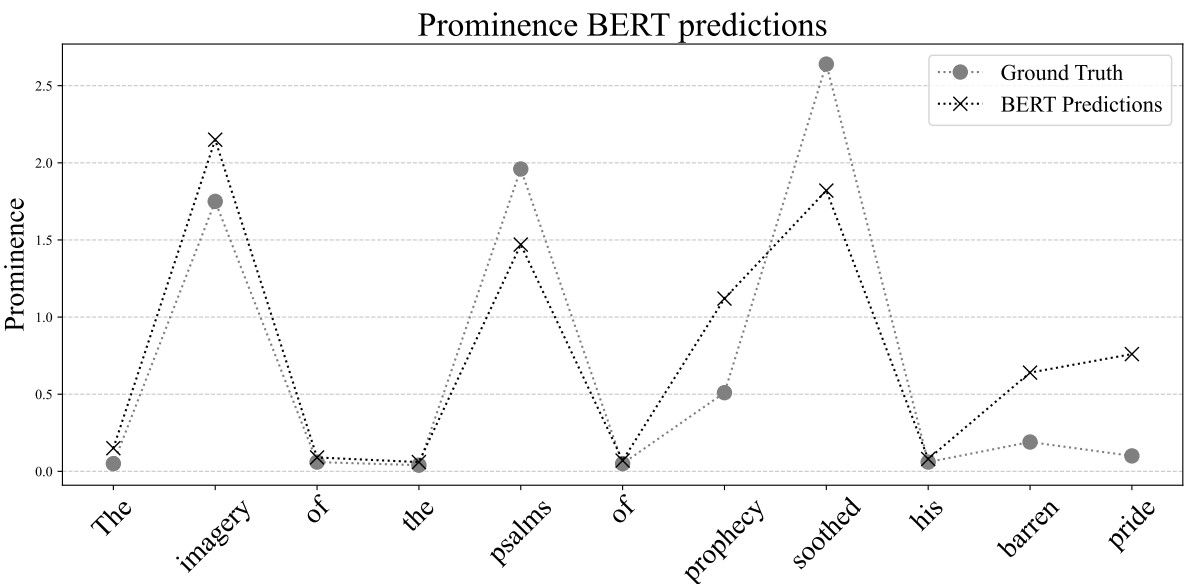

Figure 8: Prominence scores taken from Talman et al. (2019) predicted based on the shown example sentence with finetunedBERT (Devlin et al., 2019). We predict the mean of the predictive distribution parameterized by BERT.