# OpenReview forum: "Quantifying the redundancy between prosody and text"
_EMNLP/2023/Conference — EMNLP 2023 Main_

### Official Review · Reviewer_UNb6 · 2023-08-04

**Soundness:** 5

**Excitement:**

4: Strong: This paper deepens the understanding of some phenomenon or lowers the barriers to an existing research direction.

**Paper Topic And Main Contributions:**

This paper quantifies prosodic information using large language models. Estimated mutual information was used to quantify prosodic features (prominence, energy, duration, pause, pitch). According to the paper, the MI increases between prosody and text when the preceding context is considered. This is evidence of the importance of prosodic features, since context affects what is considered prosody.

**Questions For The Authors:**

Line 313: what do you mean by phonetic features? Intensity, pitch, and duration might be considered phonetic features.
Why was prominence used as a "catch-all" prosodic feature?

**Reasons To Accept:**

This paper connects state-of-the-art NLP research (LLMs) to model prosody, It presents a novel methodology that can be employed in a variety of other languages. Furthermore, the math behind the modeling is extremely clear.

**Reasons To Reject:**

Further justification for using LLMs is necessary. Why use LLMs if their interpretability is quite opaque? A simpler baseline for comparison could strengthen the paper. I am not sure how the performance of  a simpler model could be comparable to MIs, though.
Furthermore, I feel that using the word "F0" instead of "pitch" would be better, since pitch is the perceptual correlate of F0.

**Reproducibility:**

5: Could easily reproduce the results.

**Reviewer Confidence:**

4: Quite sure. I tried to check the important points carefully. It's unlikely, though conceivable, that I missed something that should affect my ratings.

**Typos Grammar Style And Presentation Improvements:**

Large language models are abbreviated as "LMs". I would recommend "LLMs".
Line 91: Unnecessary space after opening parenthesis: ( Figure 1)

---

> ### Author Rebuttal · Authors · 2023-08-29
>
> Thank you for your review and constructive feedback.
>
> Interpretability: We believe that the interpretability of LLMs is not an issue for our investigation. In order to get tight estimations on the mutual information, we are interested in the best possible predictor (see Pimentel et al. 2020). Hence, we are in need of the best possible function for this task, where a pre-trained LLM arguably provides a good initialization for fine-tuning.
>
> Baselines: We already prepared baselines that are currently excluded from the paper, e.g., predicting the average duration for the given word type. We are considering including them in the final version of the paper. To adhere to your proposal, we could add a simple n-gram-based model as an additional baseline.
>
> Prominence as “catch-all” prosodic feature: We agree with your statement and consider intensity, pitch, and duration as phonetic features. The reason prominence (the acoustic realization of stress) can be viewed as a "catch-all" prosodic feature is that it doesn't rely on just one of these phonetic features alone. Instead, prominence is typically treated as some phonological feature that has phonetic correlates in a combination of intensity, pitch, and duration. As prominence is an important feature in theoretical phonology, treating it in this work connects our contribution to the prior literature.

---

### Official Review · Reviewer_AJ81 · 2023-08-04

**Soundness:** 4

**Excitement:**

4: Strong: This paper deepens the understanding of some phenomenon or lowers the barriers to an existing research direction.

**Paper Topic And Main Contributions:**

This paper extracts prosodic features from a large corpus of spoken English and presents a pipeline for comparing the mutual-information between prosodic and word embedding features. This serves as a quantifiable measure of how much prosodic information is not predicted by or "redundant" with the text form of the words itself.

The paper is well-written, limitations cogently discussed, and the pipeline for connecting prosodic to text information is a compelling contribution in its own right.

**Reasons To Accept:**

As above.

**Reasons To Reject:**

No major reasons to reject

**Reproducibility:**

4: Could mostly reproduce the results, but there may be some variation because of sample variance or minor variations in their interpretation of the protocol or method.

**Reviewer Confidence:**

4: Quite sure. I tried to check the important points carefully. It's unlikely, though conceivable, that I missed something that should affect my ratings.

---

> ### Author Rebuttal · Authors · 2023-08-29
>
> Thank you for your review. We greatly appreciate the time and effort you've dedicated to evaluating our work. If there are any specific concerns or areas you believe need clarification, please don't hesitate to highlight them.

---

### Official Review · Reviewer_UEwE · 2023-08-11

**Soundness:** 5

**Excitement:**

5: Transformative: This paper is likely to change its subfield or computational linguistics broadly. It should be considered for a best paper award. This paper changes the current understanding of some phenomenon, shows a widely held practice to be erroneous in someway, enables a promising direction of research for a (broad or narrow) topic, or creates an exciting new technique.

**Missing References:**




**Paper Topic And Main Contributions:**

The paper explore on the interesting topic of the quantification of meaning conveyed by prosodic features.
The work is mainly motivated by study of direct mappings between acoustic features of prosody (word duration, intensity, pitch, pauses) and the quantification of the meaning of those features with respect the word itself and the preceding and future context. The authors make use of current and open-sourced LLMs in English and analyse a public available English speech corpus, LibreTTS. The density estimations of prosodic features are obtained from CEDEX database. For measuring such information information theory is deployed by estimating Mutual Information (MI) between prosodic features (aligned with the word tokens using the Montreal aligner) and their corresponding preceding and future context. For computing so, the authors fine-tune several kind of LLMs to the task of predicting prosodic features densities based on the word or word+context.





**Questions For The Authors:**

- QUESTION A: how do yo think tokenizers (annex C.2) from the different LLMs could affect the prediction of prosodic features (in the token tagging task) and the results? For example, some words could be tokenized with different number of tokens depending of the LLM.
- QUESTION B: how the model-fine tuning is performed? is only the new added head parameters estimated and the rest of the model parameters are frozen? could it be related to the big MI gap between non-contextual estimators and contextual ones, in table 1? or a reason for the big difference (2.3 times) in MI relative prominence between them?
- QUESTION C: how pauses are codified by the LLMs? is there a special token for that? several?
- QUESTION D: Why only using 100 samples points (from the 500ms analysis window) of the pitch for the DCT projection?
- QUESTION E: Is punctuation only referring to final sentence periods or is also taking into account commas and others? are LLMs tokenizers taking them into account?
- QUESTION F: in line 605, why punctuation marks are not relevant in spoken communication? Sometimes, a pause could say a lot.
- QUESTION G: Why do you think that the log energy reports higher MI differences values between the past and future context words?
Thanks a lot for your really nice study and work.


**Reasons To Accept:**

It offers a general and easy reproducible framework for future studies on prosody or other linguistic features. The paper is excellently writing, with a consistent methodology, easy to follow and with several contributions offered by the methodology itself and the drawing of the conclusions obtained by the analysis of the big corpus employed. As the authors state in the introduction, there is few work on the suprasegmental statistics information conveyed in human language communication.
The results might imply, from the technology point of view that text based synthesis could be significantly improved by using prosodic features in addition to text inputs and, from the cognitive one, that meaning in a sentence is not completely perceived till the end of such sentence.


**Reasons To Reject:**

Only English language and one domain (audiobooks) has been studied. It would be great generalize the results to several languages to generate even more powerful conclusions and specifically to free speech in human-human communcation.

**Reproducibility:**

5: Could easily reproduce the results.

**Reviewer Confidence:**

4: Quite sure. I tried to check the important points carefully. It's unlikely, though conceivable, that I missed something that should affect my ratings.

**Typos Grammar Style And Presentation Improvements:**

-The paper is overpassing the 8 pages limit from the conference, try to adjust it, please.

---

> ### Author Rebuttal · Authors · 2023-08-29
>
> Thank you for the in-depth evaluation of our work and for the constructive feedback. We especially consider your feedback on generalizing our analysis beyond the English language as a cross-linguistic study in a follow-up to this work.
>
> Regarding your questions:
>
> Question A:
> It is true that the respective tokenizers of BERT and GPT may fragment a word into unequal numbers of tokens. As we describe in Appendix C.2, the prediction task for each word-level prosodic feature is based on the last token of a word. That means, that regardless of the number of tokens this word consists of (for a certain tokenizer), the contextualized token representation will contain the information of the whole word (and more). Since we use pre-trained language models, it is fair to assume that the majority of words were part of the training data, and thus the model has learned to segment the token sequences into words. Hence, we do not think the choice of tokenizer has a large influence on the results, much less than the architecture of the model (unidirectional vs. bidirectional).
>
> Question B:
> We fine-tune the whole neural network, i.e. the pre-trained language model and the freshly initialized prediction head. Contradicting prior work (Kakouros and O’Mahony 2023, https://arxiv.org/pdf/2304.12706.pdf) we did not experience worse performance (worse generalization) by fine-tuning the whole model compared to freezing the LM and only training a new prediction head. We will discuss this in more detail in the final version of the paper.
> In order to obtain a good estimate on MI we want the best prediction performance (see Pimentel et al. 2020) and therefore went with training the whole network on the prosody prediction task.
> The function that is being optimized in our scenario (the LLMs with a new prediction head) is much more complex (many more parameters) than the ones based on the non-contextual baselines (which only train a multilayer perceptron on the frozen pre-trained word embeddings). This might influence the performance difference between contextual and non-contextual models. Nonetheless, we believe that the primary difference between non-contextual and contextual models on the relative prominence feature stems from the context alone, as a non-contextual function always predicts the same relative prominence for a specific word. A contextual function incorporates information based on prior (unidirectional) or whole (bidirectional) context. As the relative prominence random variable has much higher entropy than the absolute prominence random variable (see Table 1), we suggest that the increased variance needs additional information (contained in the context) to be predicted accurately. We believe that future work should dive deeper into this analysis by investigating this question in a more fine-grained way. For example, instead of accumulating the metrics over the whole dataset (and all words and contexts therein), one could distinguish between word types or part of speech tags and observe the differences in the predictive power of the model.
>
> Question C:
> We use the default tokenizers of the respective models, without adding a specific pause token. Therefore, there is no codification of the pauses. The LLM has to infer the pauses in the speech from the textual context. We predict the pause after each word (as before, the prediction head operates on the contextualized embedding of the last token of each word), with a label of 0.0s in case of no pause. The model regresses the pause duration in seconds.
>
> Question D:
> Thank you for this great question. We chose 100 sample points in order to upsample/downsample the variable length segments around the stress syllable to a unified length. The audio files in the LibriTTS dataset are all at a sampling rate of 24kHz, which means, given the maximum length of the segment around the stress syllable of 500ms, we have at most 12000 sampling points in the segment. This means we could upsample all (variable length) segments to a unified length of 12000 sampling points.
> We are now going to make an argument on why we believe this is not necessary: Pitch curve estimation is a noisy process, and we are interested only in the coarse curvature of the pitch curve around the stress syllable, e.g. from lower to higher frequency. Arguably, this curve is not of high order, i.e., neglecting high-frequency components of the curve, we can approximate it well by a low-order polynomial.
> Our choice of 100 sampling points within a 0.5s window leads to a spacing between sampling points of 0.005s. By the Nyquist-Shannon sampling theorem (https://en.wikipedia.org/wiki/Nyquist%E2%80%93Shannon_sampling_theorem) this means we are able to capture frequencies in the signal of up to 100Hz, which is certainly enough to capture the coarse curvature of a pitch curve around the stress syllable of a word. We may conduct additional experiments with more and fewer sampling points to cross-check this statement.
>
> Question E:
> In this work, punctuation refers to periods, commas, and others. The LLM tokenizers take them into account during training and inference. Since we extract word-level prosodic features, punctuation tokens are not mapped to valid labels (set to None). We ignore them during training and inference.
>
> Question F:
> We observe that unlike in the case of relative prominence, the prediction performance of pauses differs quite a lot between unidirectional and bidirectional models. Since in our experiments, the punctuation is always included (see Question E), a bidirectional model that predicts the pause after the word (given the embedding of the (last) token of the considered word) will contextualize the presence of the punctuation (e.g. a comma) within the word’s embedding. Hence, to analyze whether the increased prediction performance of pauses of bidirectional models stems from the bidirectional segmental context, or just from the presence of punctuation after a word, an ablation study that removes the punctuation becomes necessary.
>
> Question G:
> We have the following hypothesis on why the MI difference in energy is higher than, e.g., in pauses. While pauses are primarily indicative of structural or syntactic boundaries in speech, such as punctuation or syntactical breaks, the energy of a speech signal represents a culmination of multiple factors, including emotion, emphasis, and semantic relations. Although pauses are largely influenced by prior context (e.g., the preceding sentence structure might suggest an upcoming pause to signal the end of a thought), energy dynamics can be influenced by both prior and upcoming context. For instance, a speaker might raise their energy levels not just because of the word they are currently saying, but in anticipation of a following important or emphatic word.

---

### Meta-Review · Area_Chair_AUwe · 2023-09-19

**Recommendation:** 5

**Metareview:**

This paper uses modeling techniques to further understanding of human language and cognition. It is an excellent example of its kind.

---

### Decision · Program_Chairs · 2023-10-07

**Decision:**

Accept-Main

**Comment:**

This paper uses modeling techniques to further understanding of human language and cognition. It is an excellent example of its kind.